# Effect of Subtropical Natural Exposure on the Bond Behavior of FRP-Concrete Interface

**DOI:** 10.3390/polym12040967

**Published:** 2020-04-21

**Authors:** Xinyan Guo, Shenyunhao Shu, Yilin Wang, Peiyan Huang, Jiaxiang Lin, Yongchang Guo

**Affiliations:** 1School of Civil Engineering and Transportation, South China University of Technology, Guangzhou 510640, China; xyguo@scut.edu.cn (X.G.); ctshusyh@mail.scut.edu.cn (S.S.); 201820106463@mail.scut.edu.cn (Y.W.); 2State Key Laboratory of Subtropical Building Science, South China University of Technology, Guangzhou 510640, China; 3School of Civil and Transportation Engineering, Guangdong University of Technology, Guangzhou 510006, China; jxiang.lin@gdut.edu.cn (J.L.); guoyc@gdut.edu.cn (Y.G.)

**Keywords:** fiber reinforced polymer (FRP), bond, natural exposure, reinforced concrete (RC) beam

## Abstract

Subtropical natural exposure may significantly affect the bonding behavior of fiber reinforced polymer (FRP) externally bonded to concrete. To study the effect of subtropical natural climates on the FRP-concrete interface, natural exposure tests and an analytical approach were carried out on specimens externally bonded with carbon fiber reinforced polymer (CFRP) and basalt fiber reinforced polymer (BFRP). The bilinear bond stress-slip relationships for different exposure periods were derived from the experimental results of the strengthened reinforced concrete (RC) beams. Based on these bond-slip relationships, the full-range behavior of shear stress along the bond length and debonding load can be obtained through the analytical solution. The testing and numerical results showed that subtropical natural exposure can greatly affect the bond behavior of CFRP-concrete and BFRP-concrete interfaces in the early exposure period. In the late exposure period, the bond behavior was basically stable. With the increase of exposure time, the position of maximum shear stress tended to move backward, which indicated that the behavior of the FRP-concrete interface was weakened by natural exposure. Compared to the CFRP-concrete interface, subtropical natural exposure has greater influence on the bond behavior of the BFRP-concrete interface.

## 1. Introduction

Reinforced concrete (RC) structures can be effectively strengthened by external bonding of fiber reinforced polymer (FRP) composites for the convenience and efficiency. This technique has been demonstrated by many researchers and has become popular more than two decades years. Extensive research has revealed that bond strength of the FRP-concrete interface is crucial to the performance of FRP-strengthened concrete structures.

In recent years, more and more studies have begun to pay attention to the effect of severe environments, such as humidity, temperature, chloride, and acidic environments on FRP-strengthened concrete structures [1,2]. The environments, especially, degrade the performance of FRP-concrete interface and change its failure mechanism. Because most FRP strengthened concrete structures are often exposed in the environment to temperature and moisture, the effects of combined moisture and heat are considered by many researchers. To date, the durability of FRP-concrete interfaces under hydrothermal environments have largely been tested by accelerated ageing approaches. Tuakta [3] submerged specimens into water at temperatures of 20 °C and 50 °C for 8 weeks with several wet-dry cycles. Under moisture cyclic condition, weakening in bond strength was observed. Cromwell [4] carried out experiments considering water exposure which showed that higher temperatures (above 60 °C) may affect the bond properties, and proposed that exposure time should exceed 10,000 h. Wan [5] found that the bond behavior of the CFRP-concrete interface degraded when the CFRP bonded specimens were submerged in water for eight weeks. Ren [6] studied the bond strength between concrete and FRP after the specimens were subjected to 98% humidity at 40 °C for 1000 h. The influence of the wet-thermal environment on the CFRP-concrete interface was significant. Ramani [7] designed a series of cyclic moisture environments including 50 °C and 100% RH, and 23 °C and 35% RH, room conditions to study the effect of moisture environments on the CFRP-concrete bond. The shear joints were exposed for five weeks before testing, and the results showed that the behavior of the CFRP-concrete bond declined by 28%. Benzarti [8] carried out pull-off tests under an accelerated aging condition (40 °C, RH 95%). It was found that moisture caused a progressive decrease in the strength of the bonded interface. Zheng [9] kept the specimens in an environmental chamber for 14 days; the RH and temperature condition were set as 95% and 60 °C, respectively. The failure mode and behavior of the CFRP-concrete interface changed compared to specimens without environmental impact.

From the above literature review, it can be seen that many studies involving the interfacial durability caused by hydrothermal environment focused on the accelerated aging method [10,11,12]. However, many researchers found that material under outdoor natural exposure environments, especially in a natural subtropical climate, degraded more rapidly compared to that in indoor accelerated aging experiments [13,14,15,16]. Subtropical climates are mainly distributed in the south of China, the southeastern part of the United States and Brazil, and the coastal areas of Argentina and Australia. The most prominent feature is a hot and rainy summer. Due to the sunshine and abundant rain in subtropical climates, it was necessary to assess the degradation of the FRP-concrete interface of structures strengthened with FRP in this environmental condition.

However, the performance of FRP-concrete under outdoor subtropical exposure is still under investigation. Shukur Abu Hassan [17] conducted pull-out tests and found that after exposure to subtropical outdoor conditions for six months, the average bond strength of specimens decreased. Jovan Tatar [18] performed the FRP concrete bond durability test and compared the results of the laboratory and field conditions. Tests show that the FRP concrete bond failure mode characteristics of some field specimens changed within six months of site exposure. Hunter-Alarcóna [19] developed the aging procedure for testing single lap joints subjected to natural exposure (in Temuco, Chile) for twelve months and showed that the mechanical properties of single lap joints decreased after the natural aging process. Muhammad Ikramul Kabir [20] carried out single shear tests of CFRP concrete prisms exposed to wet-dry cycles, temperature cycles, and the outdoor environment of Sydney, Australia. Based on the results, wet-dry cycles caused significant degradation in fracture energy, whereas the initial deterioration was observed for outdoor environment.

To date, the experimental methods for studying the interfacial behavior of FRP-concrete are pull-off tests, which can be divided into three different types, as shown in Figure 1 [21,22,23]. Single-shear, double-shear, and bending tests are all pull-off tests on adhesive joints which are pulled at the end of FRP to cause debonding failure. Each of these three types of pull-off tests has its own advantages and disadvantages. The advantage of single-shear test is easy operation. However, it is rather difficult to keep the loading end horizontal, which may introduce to eccentric loading. Although the double-shear test can overcome the shortcomings of eccentric tension, it cannot simulate the actual situation of concrete structure strengthened with FRP because most concrete structures were in bending.

The objective of the present study is to propose an analytical model of bending test to describe the degradation of the FRP-concrete interface under natural subtropical climates. To better simulate the actual situation of concrete structures strengthened with FRP, bending tests were chosen and designed as shown in Figure 2 in this study. Moreover, two types of FRP, carbon fiber reinforced polymer (CFRP) and basalt fiber reinforced polymer (BFRP), were used. Considering three different exposure periods, specimens were subjected to natural exposure (in Guangzhou, China) for 0, 180, and 360 days, respectively. Moreover, bending tests were conducted to measure the strains of FRP, the relative slip between two concrete blocks, and the debonding load. Meanwhile, a deterioration model was proposed to describe the bond stress-slip relationship of CFRP-concrete and BFRP-concrete interfaces subjected to natural subtropical climates, based on the bond mechanisms. Specimens were prepared and exposed to subtropical outdoor conditions for six months and 12 months, respectively. Also, a number of tests were conducted to investigate CFRP-concrete and BFRP-concrete bonding performance. The proposed solutions that estimate the critical load were also reported on and compared with test results.

## 2. The Proposed Model

The structural behavior of bending test is more complex than that of single-shear and double-shear tests. To find the bond-slip relationship, the stress distribution, and the slip along the length of FRP-concrete bond interface, the differential equations of the interfacial slip and the solution of the differential equations in different stages were set up in this section.

### 2.1. Differential Equations of the Interfacial Slip

The beam in Figure 2 was subjected to two symmetrical loads. Based on the force analysis of the specimens designed in this paper between the two load points, the shear force on the concrete is zero, and the bending moment is mainly afforded by the steel bar.

Some assumptions were provided before establishing the differential equations: the beam weight has no effect on the interfacial stress; the behavior of three components (concrete, adhesive layer, and FRP plate) are linear elastic; the cross section of beam remain plane; the adhesive layer is subjected to shear deformation, and FRP is assumed to experience tensile deformation; and the thickness and width of concrete, adhesive layer, and FRP plate are constant. Based on these assumptions, the slip and stress of the FRP-concrete interface can be derived by the following equations:

The horizontal equilibrium equations of the FRP-concrete interface were set up as follows:(1)$$\tau \left(x\right)dx=d\sigma_{p}\left(x\right)t_{p}$$
(2)$$\frac{d\sigma_{c}\left(x\right)}{dx}+\frac{\tau \left(x\right)\cdot b_{p}}{b_{c}\cdot h}=0$$
where τ(x) is interfacial shear stress in the adhesive layer; $\sigma_{c}\left(x\right)$ is normal stress in the concrete; $\sigma_{p}\left(x\right)$ is tensile stress on FRP; $t_{p}$ and $b_{p}$ are the height and width of FRP, respectively; h and $b_{c}$ are the height and width of concrete, respectively. The constitutive equations for the concrete, adhesive layer, and plate are described by:(3)$$\sigma_{c}\left(x\right)=E_{c}\frac{du_{c}\left(x\right)}{dx}$$
(4)τ=τ(δ)
(5)$$\sigma_{p}\left(x\right)=E_{p}\frac{du_{p}\left(x\right)}{dx}$$
where $u_{c}\left(x\right)$ and $u_{p}\left(x\right)$ are longitudinal displacements of the concrete and the FRP, respectively. $E_{c}$ and $E_{p}$ are elastic modulus of concrete and FRP, respectively. δ(x) is the interfacial slip between the FRP and concrete, which is determined by:(6)$$\delta \left(x\right)=u_{p}\left(x\right)-u_{c}\left(x\right)$$

From Equations (1)–(6), the second order differential equation can be obtained as follows:(7)$$\frac{d^{2}\delta \left(x\right)}{dx^{2}}=\lambda^{2}\tau \left(\delta \right)$$
where, $\lambda^{2}=\frac{1}{E_{p}\cdot t_{p}}+\frac{b_{p}}{E_{c}\cdot b_{c}\cdot h}$.

The beam in Figure 2 was subjected to two symmetrical loads. In boundary conditions, the tensile stress of FRP at the original location (*x* = 0) shown in Figure 2 can be derived based on equilibrium equations on cross section:(8)$$\sigma_{p0}A_{p}+\sigma_{st0}A_{st}=0$$
(9)$$\sigma_{p0}A_{p}y+\sigma_{st0}A_{st}\;\left(h-y-c\right)=M$$
where $\sigma_{p0}$ and $\sigma_{st0}$ are the tensile stresses on FRP and steel bar at *x* = 0, respectively; $A_{p}$ is the area of FRP, $A_{st}$ is the area of the steel bar; y is the height of the neutral axis of the cross section from the bottom of beam; and c is the distance between the steel bar and the top of the beam (shown in Figure 2), respectively. With the Equations (8) and (9), the tensile stress of FRP at *x* = 0 is obtained by:(10)$$\sigma_{p0}=\frac{M}{\left(h-c\right)A_{p}}$$

### 2.2. Natural Exposure Effect on Bond-Slip Models

The bond-slip relation is a vital parameter to describe the interfacial behavior of FRP-concrete interface. So far, six types of available bond-slip models for FRP-concrete interfaces have been provided as follows: (a) elastic-plastic model, (b) cutoff model, (c) bilinear model, (d) Popovics’ model [24], (e) third-order polynomial model [25,26], (f) exponential softening model [27], and (g) fracture energy model [28]. The elastoplastic type and cutoff type should be carefully used because they do not consider slip after the maximum bond stress. Both the bilinear and Popovics’ type have an ascending and descending branch. Moreover, Popovics’ type is a numerical approach similar to concrete stress-strain relationship, but the expression of stress-slip is complex so it is difficult to substitute into a differential equation to find the stress distribution and slip at certain load. In this study, the bilinear type model was used.

The bilinear bond-slip relationship is composed of an initial elastic ascending branch, descending branch, and debonding stage. Bond shear stress increases linearly until it reaches the maximum bond stress, $\tau_{f}$ (when the relative slip $\delta_{1}$ corresponding to $\tau_{f}$), at the ascending branch; while decreases linearly until it becomes zero (when the relative slip $\delta_{f}$ corresponding to zero stress), at the descending branch. At debonding stage, the shear stress is zero. The bond-slip relationship is shown in Figure 3 and expressed by the following equation:(11)$$\tau \left(\delta \right)=\{\begin{array}{l} \frac{\tau_{f}}{\delta_{1}}\delta ;\;0<\delta \leq \delta_{1}\\ \frac{\tau_{f}}{\delta_{f}-\delta_{1}}\left(\delta_{f}-\delta \right);\;\delta_{1}<\delta \leq \delta_{f}\\ 0;\;\delta >\delta_{f} \end{array}$$

In the model, the ascending branch and descending branch are characterized by three values: maximum bond stress ($\tau_{f}$), the slip ($\delta_{1}$) corresponding to maximum stress ($\tau_{f}$), as well as maximum slip ($\delta_{f}$) at zero stress.

Although several models [25,26,27,28,29,30,31,32,33,34,35] were proposed considering the effects of aggressive environments (such as sulphate attack, temperature change, or wet-dry cycles) on interfacial behavior, no direct models were provided for FRP-concrete interfaces after exposure to natural subtropical climates. In some models [25,26,29,30,34], the maximum stress ($\tau_{f}$) and the slip ($\delta_{1}$) corresponding to maximum stress were revised. In this paper, three values (maximum bond stress, $\tau_{f}$; the slip, $\delta_{1}$, corresponding to maximum stress, $\tau_{f}$; and maximum slip, $\delta_{f}$, at zero stress) were revised as $\tau_{f}\left(n\right)$, $\delta_{1}\left(n\right)$, and $\delta_{f}\left(n\right)$ based on the experimental data. The subtropical natural exposure tests were carried out in Section 3.

### 2.3. Theoretical Results in Different Stages

From Equations (7) and (11), the differential equation can be solved to obtain the interfacial slip, interfacial shear stress distribution along the adhesive interface, and the ultimate load under the effect of natural subtropical climate exposure. As shown in the bond-slip relationship, the behavior of FRP-concrete interfaces can be divided into the ascending stage, descending stage, and debonding stage.

#### 2.3.1. Ascending Stage

This stage is the elastic stage, during which the interfacial shear stress is less than maximum stress, $\tau_{f}$. At this stage, with the load increase (a→b→c→d) shown in Figure 4, the interfacial shear stresses at different locations, *x*, increases. Figure 4 is a schematic diagram made to show the interfacial shear stress increasement. Considering the effect of natural subtropical climate exposure, the differential equation is obtained:(12)$$\frac{d^{2}\delta \left(x\right)}{dx^{2}}-\lambda^{2}k_{1}\left(n\right)\delta =0$$
where $k_{1}\left(n\right)=\frac{\tau_{f}\left(n\right)}{\delta_{1}\left(n\right)}$ is the slop of bond-slip line in the ascending stage under different exposure periods, n.

The expressions for the slip and shear stress of the FRP-concrete interface and the tensile stress in FRP plate are derived:(13)$$\delta \left(x\right)=C_{1}^{\prime }e^{\lambda x\sqrt{k_{1}\left(n\right)}}+C_{2}^{\prime }e^{-\lambda x\sqrt{k_{1}\left(n\right)}}$$
(14)$$\tau \left(x\right)=k_{1}\left(n\right)\cdot \left(C_{1}^{\prime }e^{\lambda x\sqrt{k_{1}\left(n\right)}}+C_{2}^{\prime }e^{-\lambda x\sqrt{k_{1}\left(n\right)}}\right)$$
(15)$$\sigma_{p}\left(x\right)=\frac{\sqrt{k_{1}\left(n\right)}}{\lambda t_{p}}\cdot \left(C_{1}^{\prime }e^{\lambda x\sqrt{k_{1}\left(n\right)}}-C_{2}^{\prime }e^{-\lambda x\sqrt{k_{1}\left(n\right)}}\right)+C_{3}^{\prime }$$

The boundary conditions $(\sigma_{p}\left(0\right)=\frac{M}{\left(h-c\right)A_{p}}$; $\sigma_{p}\left(L_{p}\right)=0;\tau \left(0\right)=0)$ are substituted in Equations (13)–(15), and the integral constants are determined as follows:(16)$$C_{1}^{\prime }=C_{2}^{\prime }=\frac{-\lambda t_{p}\cdot \sigma_{p0}\cdot e^{\lambda L_{p}\sqrt{k_{1}\left(n\right)}}}{\sqrt{k_{1}\left(n\right)}\cdot \left(e^{2\lambda L_{p}\sqrt{k_{1}\left(n\right)}}-1\right)}$$
(17)$$C_{3}^{\prime }=\sigma_{p0}$$

When the interfacial shear stresses at the loaded end (x=0) reach $\tau_{f}\left(n\right)$, the maximum shear stress under *n* days of subtropical natural exposure, the elastic stage ends, and the bond-slip of FRP-concrete interface from the beginning location (x=0) enters the descending stage. Substituting $x=0\;\mathrm{and}\;\tau_{f}\left(n\right)$ into Equations (14), (16) and (17), the load at the end of elastic stage was derived:(18)$$\frac{M}{\left(h-c\right)A_{p}}=\frac{\tau_{f}\left(n\right)\cdot \left(1-e^{2\lambda L_{p}\sqrt{k_{1}\left(n\right)}}\right)}{2\sqrt{k_{1}\left(n\right)}\cdot \lambda L_{p}\cdot e^{\lambda L_{p}\sqrt{k_{1}\left(n\right)}}}$$

#### 2.3.2. Descending Stage

This stage is the elastic softening stage, during which the shear stress decreases, and slip increases but smaller than $\delta_{f}$. With the load increasing (a→b→c→d) shown in Figure 5, the interfacial shear stress at the loaded end, x=0, decreased and the location of the maximum bond stress, $\tau_{f}$, moved to new location. Figure 5 is a schematic diagram made to show the interfacial shear stress distribution. The zone from location x=0 to the new location is called the softening zone, and the length of the zone is denoted by *s*, which increased as the load increased.

Considering the effect of natural subtropical climate exposure, the differential equation can be obtained:(19)$$\frac{d^{2}\delta \left(x\right)}{dx^{2}}+\lambda^{2}k_{2}\left(n\right)\delta -\lambda^{2}k_{2}\left(n\right)\delta_{f}\left(n\right)=0$$
where, $k_{2}\left(n\right)=\frac{\tau_{f}\left(n\right)}{\delta_{f}\left(n\right)-\delta_{1}\left(n\right)}$ is the slop of bond-slip line in the descending stage under different exposure periods, n.

The expressions for the slip and shear stress of the FRP-concrete interface and the tensile stress in FRP plate are derived:(20)$$\delta \left(x\right)=C_{1}^{\prime \prime }\cos[\lambda x\sqrt{k_{2}\left(n\right)}]+C_{2}^{\prime \prime }\sin\left[\lambda x\sqrt{k_{2}\left(n\right)}\right]+\delta_{f}\left(n\right)$$
(21)$$\tau \left(x\right)=-k_{2}\left(n\right)\cdot \left(C_{1}^{\prime \prime }\cos\left[\lambda x\sqrt{k_{2}\left(n\right)}\right]+C_{2}^{\prime \prime }\sin\left[\lambda x\sqrt{k_{2}\left(n\right)}\right]\right)$$
(22)$$\sigma_{p}\left(x\right)=\frac{\sqrt{k_{2}\left(n\right)}}{\lambda t_{p}}\cdot \left(-C_{1}^{\prime \prime }\sin\left[\lambda x\sqrt{k_{2}\left(n\right)}\right]+C_{2}^{\prime \prime }\cos\left[\lambda x\sqrt{k_{2}\left(n\right)}\right]\right)+C_{3}^{\prime \prime }$$

The boundary conditions are listed as:$$\sigma_{p}\left(0\right)=\frac{M}{\left(h-c\right)A_{p}};\;\sigma_{p}\left(L_{p}\right)=0;$$

Besides the above boundary conditions, the tensile stress of FRP at the maximum interfacial shear stress is continuous. According to the continuity condition, $\sigma_{p}\left(s\right)$ calculated by Equation (15) is equal to $\sigma_{p}\left(s\right)$ calculated by Equation (22). In the descending stage, the integral constants ($C_{1}^{\prime \prime }$, $C_{2}^{\prime \prime }$, $C_{3}^{\prime \prime }$) and *s* were determined based on systems of equations as follows:(23)$$\{\begin{array}{l} \frac{\sqrt{k_{2}\left(n\right)}}{\lambda t_{p}}\cdot C_{2}^{\prime \prime }+C_{3}^{\prime \prime }=\sigma_{p0}\\ \frac{\sqrt{k_{2}\left(n\right)}}{\lambda t_{p}}\cdot \left(C_{1}^{\prime \prime }\sin\left[\lambda L_{p}\sqrt{k_{2}\left(n\right)}\right]+C_{2}^{\prime \prime }\cos\left[\lambda L_{p}\sqrt{k_{2}\left(n\right)}\right]\right)+C_{3}^{\prime \prime }=0\\ -k_{2}\left(n\right)\cdot \left(C_{1}^{\prime \prime }\cos\left[\lambda s\sqrt{k_{2}\left(n\right)}\right]+C_{2}^{\prime \prime }\sin\left[\lambda s\sqrt{k_{2}\left(n\right)}\right]\right)=\tau_{f}\left(n\right)\\ \frac{\sqrt{k_{1}\left(n\right)}}{\lambda t_{p}}\cdot \left(C_{1}^{\prime }e^{\lambda s\sqrt{k_{1}\left(n\right)}}-C_{2}^{\prime }e^{-\lambda s\sqrt{k_{1}\left(n\right)}}\right)+C_{3}^{\prime }=\\ \frac{\sqrt{k_{2}\left(n\right)}}{\lambda t_{p}}\cdot \left(-C_{1}^{\prime \prime }\sin\left[\lambda L_{p}\sqrt{k_{2}\left(n\right)}\right]+C_{2}^{\prime \prime }\cos\left[\lambda s\sqrt{k_{2}\left(n\right)}\right]\right)+C_{3}^{\prime \prime } \end{array}$$

With the Equations (20)–(22), interfacial slip, interfacial shear stress, tensile stress in FRP plate, and the length of the softening zone of the FRP-concrete interface in the descending stage can be calculated.

At the end of this stage, the interfacial slip at x=0 reaches maximum slip, $\delta_{f}\left(n\right)$. Substituting the condition $\delta \left(0\right)=\delta_{f}\left(n\right)$ into Equation (20), the debonding load, $P_{u}$, can be derived. $P_{u}$ is debonding load, and the criterion of debonding failure is the that the slip reaches the maximum slip, $\delta_{f}\left(n\right)$.

#### 2.3.3. Debonding Stage

At the beginning of this stage, the interfacial slip at x=0 reaches maximum slip $\delta_{f}\left(n\right)$ and the interfacial shear stress at this location decreased to zero, then the debonding crack occurred. As the load increased (a→b→c→d) shown in Figure 6, the debonding crack propagated and the location of zero shear stress moves away along the interface from x=0. In this stage, most of the interfacial shear stress began to decrease. Interfacial slip, interfacial shear stress, and tensile stress in FRP plate of the intact interface can still be obtained by Equations (20)–(22). Figure 6 is a schematic diagram made to show the regular of interfacial shear stress distribution.

## 3. Experimental Program

### 3.1. Exposure Process

The exposure process of the specimens was performed in a natural subtropical environment in Guangdong Province, China. The most prominent feature of this area is a hot and rainy summer. The annual temperature of the natural environment and concrete structures are shown in Figure 7. In the geographical area, the average annual temperature is 21.6–22.3 °C; the highest temperature in summer reach around 40 °C; the highest temperature on concrete structures, such as a bridge, was over 50 °C; and the temperature differences between day and night is about 10 °C [36].

Figure 8 shows the annual average relative humidity of the natural subtropical environment and concrete structures. During the exposure process, the average annual relative humidity reaches 78%. In addition, the highest relative humidity of concrete structures reaches 98% [37].

### 3.2. Test Specimens

In this study, a total of six specimens were tested. The tests were carried out with two variables: (1) types of FRP (basalt, carbon) and (2) exposure periods (0, 180, and 360 days). Figure 9 shows the geometrical dimensions of specimens. Each specimen consisted of two 300 mm long concrete blocks with a cross section of 100 mm × 150 mm. Two concrete blocks were connected by two HRB hot-rolled ribbed bars of 650 mm in length and 12 mm in diameter. A 10 mm gap was reserved between the two concrete blocks. According to the various effective bond lengths mentioned [16], the maximum effective length of the specimen was 109 mm. Therefore, the bond length of FRP was supposed as 150 mm. FRP sheets 50 mm wide and 410 mm long were externally bonded to the bottom of the specimen. To ensure the debonding failure occurs first on one side, shear strengthening was performed along the other side by wrapping the 200 mm wide and 600 mm length FRP sheet.

### 3.3. Material Details

In this study, FRP used in this paper were CFRP and BFRP. CFRP sheets were UT70-30 and BFRP sheets were produced by Zhejiang Shijin Basalt Fiber Co., Ltd. The adhesive used for bonding FRP to concrete was epoxy adhesive. The properties of the CFRP, BFRP, and resin epoxy were given by the manufacturer and are listed in Table 1.

The concrete prepared for all specimens was ordinary strength concrete cast in the lab. The weight ratio of the main ingredients of concrete was cement (1.0): water (0.64): sand (2.02): gravel (3.59). Three cylindrical test blocks were tested, and the average compressive strength after 28 days of curing was 30.5 MPa. The characteristics of materials used in the test are summarized in Table 1.

### 3.4. Test Set-Up

Four-point bending experiments were carried out in this paper and all specimens were loaded on a hydraulic machine of 500 kN capacity. Displacement-controlled monotonic loading history was adopted, and the loading was applied by a two-point loading system. The distance between the two loading points and two supports were 100 mm and 310 mm, respectively. Monotonic loading was applied with a constant rate of 0.2 mm/min. Force sensors were installed on both load cells to record applied loads. During each test, the linearly variable differential transducer (LVDT), with a range of 50 mm and an accuracy of 0.01 mm, was positioned at the mid-span to measure the deflection of specimens.

To record the strains on FRPs, a total of eight strain gauges were externally bonded along the length of FRP at an interval distance of 20 mm, as shown in Figure 10. Strains were automatically recorded by static data acquisition instruments during the whole experimental process. Specimens were preloaded before formal loading, to ensure good machine condition and eliminate gaps between the load cell and specimens.

## 4. Test results

The results from the bending tests on specimens for bond behavior between FRP-concrete are presented, while considering the effect of type of FRP and exposure time. Description of ultimate loads, failure modes, and strains of FRP under different loads are reported.

### 4.1. Debonding Loads

Table 2 shows the debonding loads of specimens after exposure to a natural subtropical environment for 0, 180, and 360 days. Specimens marked 0 days are the reference beams without aging. Inspection of Table 2 reveals that ultimate loads of specimens strengthened with different types of FRP decreased rapidly before exposure for 180 days, while specimens with CFRP and BFRP decreased by 13.6% and 26.9%, respectively. After 180 days of exposure, the debonding loads tend to be stable.

### 4.2. Failure Mode

Subjected to natural exposure in the environment, the concrete surface of the specimen turned white, and FRP sheet was light green compared to reference. There is no significant difference in the surface of the specimens with exposure period of 180 days and 360 days.

The failure mode of most specimens was debonding of the FRP-concrete interface, and the concrete with a different thickness was attached to FRP. At the beginning of the failure process, there was a subtle sound at the mid-span of the beam. As the loading increased, some oblique cracks began to appear on the surface of the concrete with popping sounds and finally resulted in debonding of FRP from concrete. For reference specimens without aging, the failure mode of specimens with CFRP was concrete shear followed by debonding failure, while the specimens with BFRP occurred with debonding failure. The failure mode of specimens subjected to natural exposure in the environment was debonding failure on FRP-concrete interfaces. The failure modes of all specimens are shown in Figure 11, which depicts that the bonding behavior of CFRP-concrete is better than that of BFRP-concrete. With the prolonged exposure time, the bonding behavior of CFRP-concrete and BFRP-concrete decreased.

### 4.3. Strains on CFRP and BFRP

The effect of exposure time on bond behavior of the FRP-concrete interface can be described through Figure 12, which shows the strains on CFRP and BFRP along the bond length under different loads. The horizontal coordinate is the distance from the strain gauge to the vicinity of the mid-span, and the vertical coordinate is the strain value.

At the beginning of loading, only the first and second strain gauge near the mid-span had strain value. Moreover, the value of the first strain gauge nearest the mid-span was much larger than the second strain gauge. As the load increases, the strains on FRP at far distances began to appear, and the strains near the mid-span increased dramatically. When the FRP debonded at the mid-span, the interface stress was rearranged and the strain on far distance reached maximum value, followed by the fully debonding of FRP from the concrete surface. As the applied load was close to the ultimate load, the value of the strain gauge far from the mid-span was small, indicating that the ultimate load of specimens would not increase with the increase of bond length.

The results of Figure 12 indicate similar characteristics in the strain distribution curve of all specimens. As the exposure period increases, the effect of exposure period on bond behavior shows as follows: The position of the peak strain moves towards the free end of FRP. The observations indicated that natural exposure to the environment degraded the bond behavior of FRP-concrete interface.

### 4.4. Bond-Slip Relationship

Based on the data of Figure 12, the fitting double lines of bond stress-slip relationships of specimens under different exposure periods are shown in Figure 13. The shear stress and slip of all points in Figure 13 were calculated from the strain values on the FRP strain under different loads.

The average bond stress, τ, between two strain gauges can be obtained from the following equation:(24)$$\tau =E_{p}t_{p}\frac{\varepsilon_{i}-\varepsilon_{i-1}}{\Delta x}$$
where $\varepsilon_{i}$ and $\varepsilon_{i-1}$ are strain values of two adjacent strain gauges, and Δx is the distance of two adjacent strain gauges.

The slip, δ, at point $x_{i}$ can be obtained from the following equation:(25)$$\delta \left(x_{i}\right)=\delta \left(x_{i+1}\right)+\int_{x_{i+1}}^{x_{i}}\varepsilon \left(x\right)dx$$

These points were fitted to the ascending and descending stages to obtain the bond-slip relationship between the CFRP-concrete interface and the BFRP-concrete interface after different exposure periods. The major bond parameters for bond-slip relationship are listed in Table 3.

Figure 13 and Table 3 display the effects of natural subtropical exposure on FRP-concrete bond behavior. Comparing the results of specimens after exposure periods with reference beams without exposure, the similarities of CFRP-concrete and BFRP-concrete interface were described as follows:◆The maximum bond shear stress, $\tau_{f}$, experienced a significant decrement at the early exposure period (0–180 days), and the reductions of 26.8% and 31.9% were found for CFRP-concrete and BFRP-concrete interface, respectively. Moreover, $\tau_{f}$ was basically stable in the late exposure period (180–360 days) and slightly reduced (9.9% and 9.1%) with the duration of exposure.◆The slip, $\delta_{1}$, corresponding to the maximum bond shear stress increased slightly during the early exposure period (0–180 days). In the later period of exposure (180–360 days), the decrease of $\delta_{1}$ was larger than that at the early exposure period, which were 6.6% and 14.5%, respectively.◆Whether for the CFRP-concrete interface or BFRP-concrete interface, the exposure time had a great influence on the interface stiffness. As the exposure time increased, the interface stiffness degraded rapidly in the ascending and descending stages. During the ascending stage, reductions of 18.9% and 36.8% were observed for interface stiffness of CFRP-concrete and BFRP-concrete in the early exposure period, while they decreased 15.4% and 8.4% in the late exposure period. The results showed that the early stage of exposure had a greater effect on stiffness degradation. Otherwise, similar trends were observed in the descending stage, in which the early stage of exposure had a greater effect on stiffness degradation and the maximum degradation reached 37.8%.

The difference of two different type interfaces was that bond behavior of the BFRP-concrete interface was weaker than that of the CFRP-concrete interface, which was mainly characterized by smaller maximum shear stress, larger slip, and poorer stiffness. Figure 13 displays the data of the BFRP-concrete interface has large dispersion, which shows the instability of the mechanical properties of the interface.

## 5. Analytical Model and Validation

In terms to test results, the bond-slip relationships of FRP-concrete exposed to natural subtropical environments were derived in Section 3 and Section 4. Substituting these bond-slip relationships into the proposed model in Section 2, shear stress distribution of CFRP-concrete and BFRP-concrete interfaces under different loads can be evaluated. Moreover, when the maximum slip was used as the failure criterion, the debonding load of specimens exposed to different exposure times can be derived.

To validate the proposed model, its predictions were compared to the experimental results. When the predictions were in good agreement with the experimental results, the proposed model provided an analytical approach to obtain the bond shear stress distribution of FRP-concrete and debonding load, in terms of the bond stress-slip relationship of the FRP-concrete interface.

### 5.1. Computing Procedure

Based on the bond stress-slip relationship of FRP-concrete interfaces exposed to different exposure times, the bond shear stress distribution of FRP-concrete interface and debonding load can be determined by the following computing procedure. The procedure is outlined in the flow chart in Figure 14.◆Input some major parameters of a certain type of bond stress-slip relationship, such as $\tau_{f}\left(n\right)$, $\delta_{1}\left(n\right)$, $\delta_{f}\left(n\right)$, $k_{1}\left(n\right),\;\mathrm{and}\;k_{2}\left(n\right)$. For different exposure times, these parameters were different.◆Take a small load as the initial load and set the load increment, ΔP. The major parameters are substituted into Equations (15)–(17) in the ascending stage to obtain τ(x), δ(x), and $\sigma_{p}\left(x\right)$ under the initial load. With the increase of the load, according to the output values of τ(x) and δ(x), it can be judged whether the stress-slip relationship at the *x* position was in the ascending or descending stage. If it is still in ascending stage, then the procedure enters cycle (1), otherwise it enters cycle (2).◆After entering cycle (2), cycle (1) stops and Equations (20)–(23) are used in the descending stage.◆When the slip at the *x* position reaches $\delta_{f}\left(n\right)$, the FRP at *x* position is debonded. The shear stress is rearranged, and the procedure entered into the debonding stage. In this stage, the debonding load is obtained.

### 5.2. Exposure Time Effect on Stress Distribution along the FRP-Concrete Interface

Figure 15 presents the comparison of an analytical solution and test results of specimen NE-CFRP-180. The comparison results showed that the full-range interfacial shear stress distribution of both results was similar. The interfacial shear stress experienced ascending and descending stages with the load increased. The differences observed in Figure 15 were the transfer rate. Compared to the test results, the analytical solutions were more uniform away from the place where the load was applied, which was mainly caused by the continuity of the analytical model.

Every bond-slip relationship of specimens under different exposure periods (0, 180, and 360 days) was substituted into the computing procedure. The shear stress distribution along the bonding length under different loads are illustrated in Figure 16.

It can be seen from the figure, that at the initial stage of loading, the bond shear stress of the CFRP-concrete and BFRP-concrete interface near the mid-span was large, while the shear stress in other regions was almost zero. It indicated that the shear interface stresses were mainly focused on the mid-span and do not transfer to the far-end. Furthermore, as the load slowly increased, the position where the maximum shear stress appeared moved backward, and the position at the far end began to show shear stress. The shear stress near mid-span reached zero first and debonding occurred. The shear stress distribution was rearranged, and shear stress near deboning zone increased rapidly.

Moreover, the transferred shear stresses throughout the FRP under different exposure periods also showed that with the increase of exposure time, under the same load, the position of maximum shear stress tended to move backward. It indicated the behavior of the FRP-concrete interface was weakened by natural exposure.

### 5.3. Debonding Load

The theoretical calculations and experimental values of debonding load under different exposure periods are compared in Figure 17, which revealed that the two results were in good agreement, and the theoretical calculations overestimated the test experimental results by a maximum of 10.5%. The reason for the higher theoretical calculations was that due to the continuity of analytical model and the uniformity of the interfacial stress, distribution was more uniform. Moreover, the tensile fracture of concrete was not taken into account in the analytical model. Compared to the references, the debonding load of specimens bonded with CFRP and BFRP after exposure for 180 days decreased by 15.2% and 31.0%, respectively.

## 6. Conclusions

Bond strength of the FRP-concrete interface is crucial to the performance of FRP reinforcement of concrete structures, and the natural subtropical climate will degrade the bond behavior. To study the effect of natural exposure on the behavior of CFRP-concrete and BFRP-concrete interface, concrete specimens bonded with CFRP and BFRP were exposed to a natural subtropical climate for 0, 180, and 360 days. Test results and analytical solutions for the full-range behavior of interfaces were presented in this paper. The findings are novel, and the main conclusions are summarized as follows:A bilinear bond stress-slip relationship of CFRP-concrete and BFRP-concrete interfaces after different natural exposure periods were obtained through experimental dates. Major parameters of the bond-slip relationship showed that the bond behavior was greatly affected in the early exposure period and was basically stable in the late.Compared to reference beam without exposure, the maximum shear stress ($\tau_{max}$), the slip ($\delta_{1}$) corresponding to maximum shear stress, and stiffness of FRP-concrete interface exposed to natural subtropical climate degraded. Among them, maximum shear stress and stiffness were most affected and had a maximum reduction of 31.9% and 36.8%, respectively.Bond behavior of the BFRP-concrete interface was weaker than that of the CFRP-concrete interface, which was mainly characterized by small maximum shear stress, large slip, and poor stiffness.The analytical model presented was versatile, because once the bond behavior of FRP-concrete was inputted into structures strengthened with FRP, the full range behavior of FRP-concrete under different loads can be evaluated and the debonding load of specimens can be derived.With the increase of exposure time, the position of maximum shear stress tended to move backward, which indicated that natural exposure can impair the behavior of the FRP-concrete interface.

## Figures and Tables

**Figure 1 polymers-12-00967-f001:**
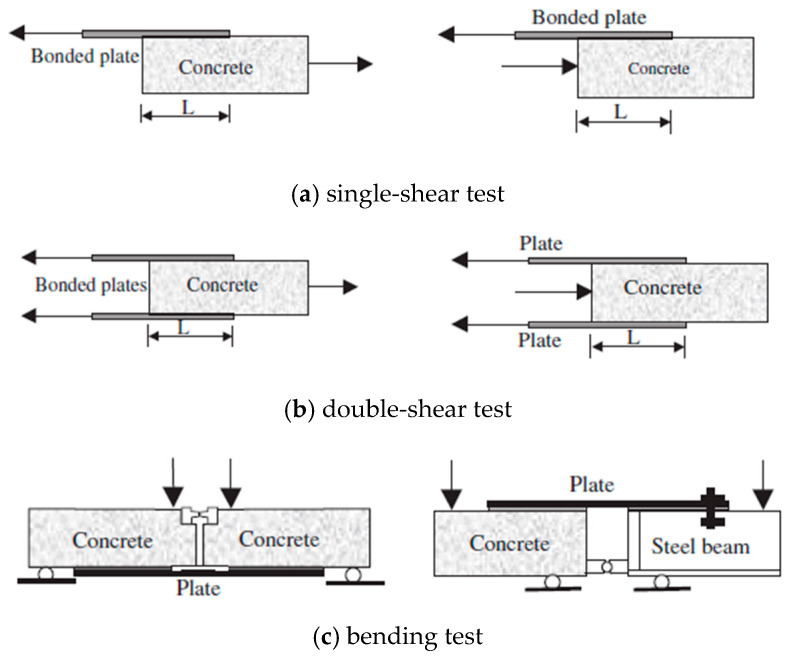
Three types of bond tests: (**a**) single-shear test; (**b**) double-shear test; and (**c**) bending test.

**Figure 2 polymers-12-00967-f002:**
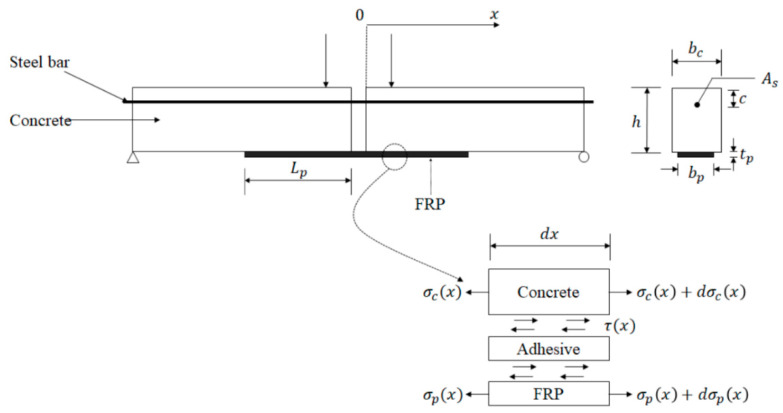
Schematic diagram of the FRP-concrete interface of bending test.

**Figure 3 polymers-12-00967-f003:**
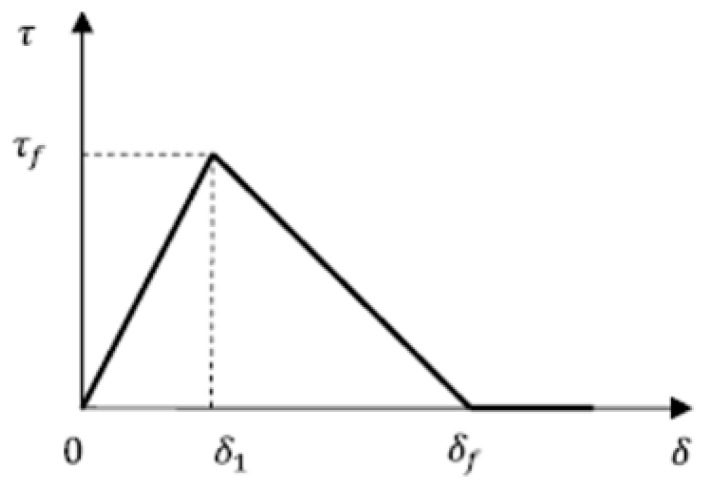
Bilinear bond-slip model.

**Figure 4 polymers-12-00967-f004:**
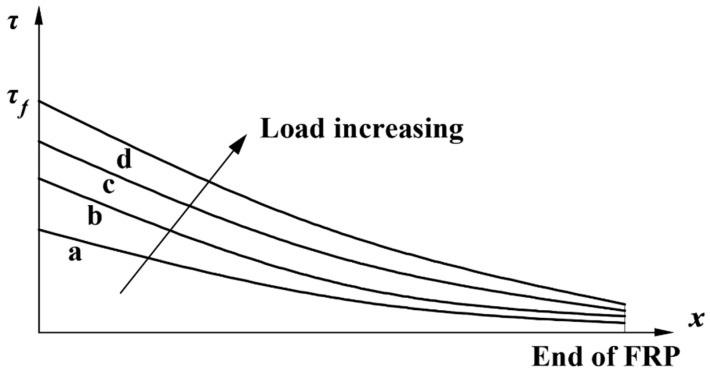
Interfacial shear stress changes at ascending stage.

**Figure 5 polymers-12-00967-f005:**
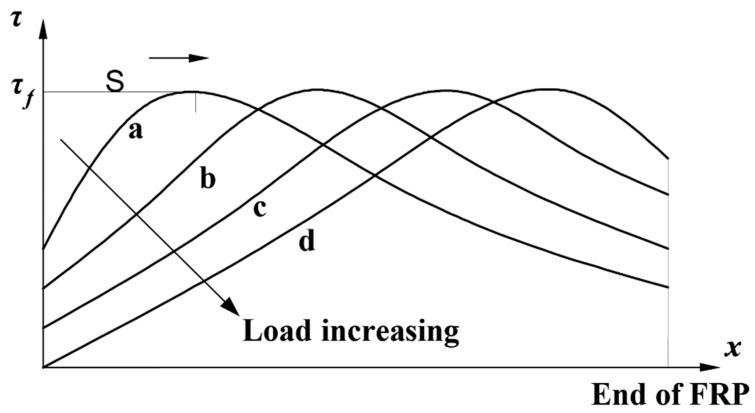
Interfacial shear stress changes at descending stage.

**Figure 6 polymers-12-00967-f006:**
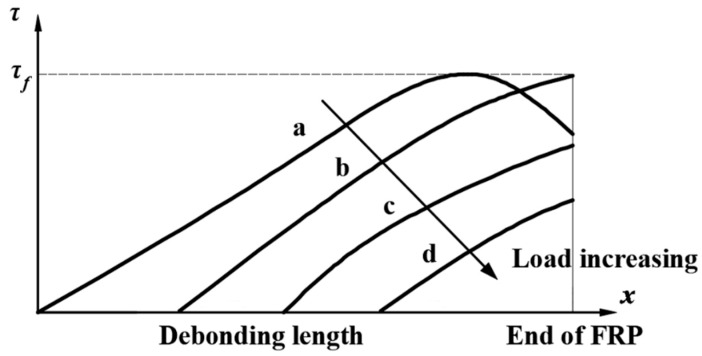
Interfacial shear stress changes at debonding stage.

**Figure 7 polymers-12-00967-f007:**
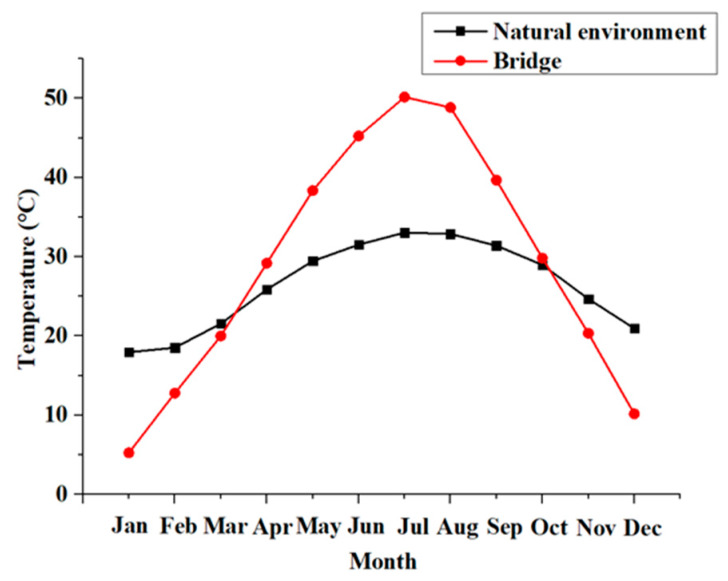
Annual temperature cycle.

**Figure 8 polymers-12-00967-f008:**
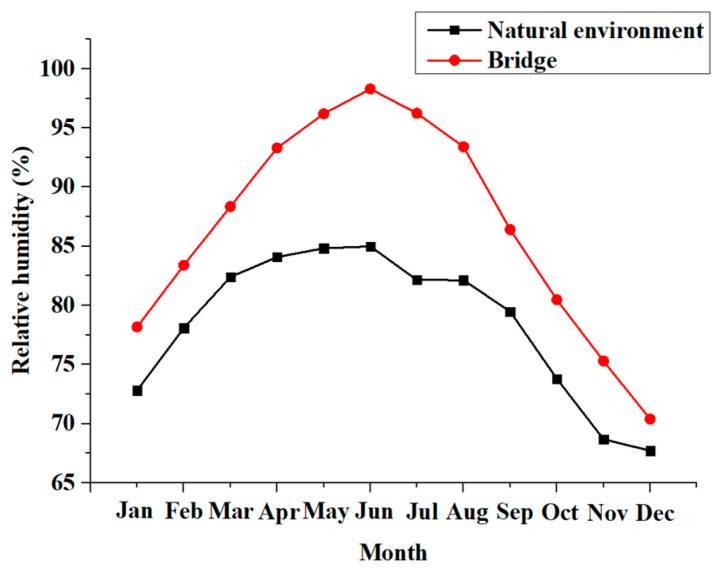
Annual humidity cycle.

**Figure 9 polymers-12-00967-f009:**
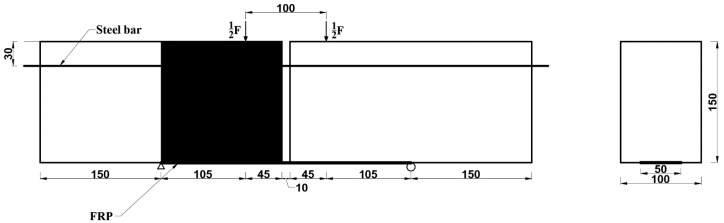
Geometric dimensions of the specimens (Unit: mm).

**Figure 10 polymers-12-00967-f010:**
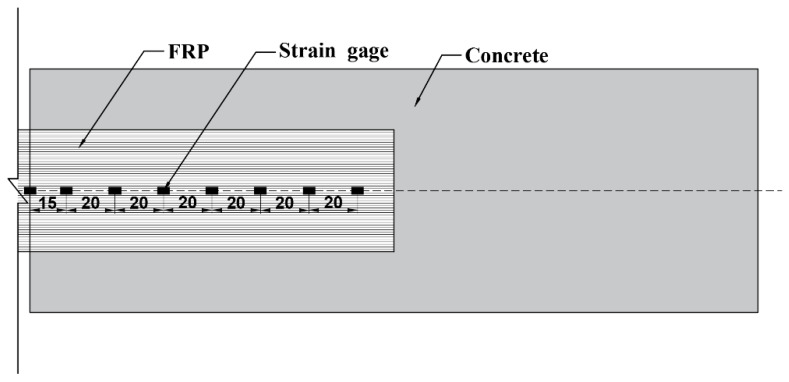
Location of the strain gauges.

**Figure 11 polymers-12-00967-f011:**
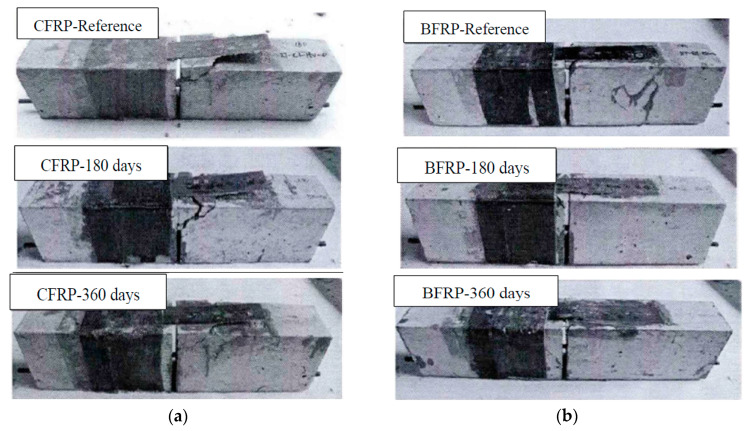
Failure modes of specimens. (**a**) Specimens with CFRP and (**b**) specimens with BFRP.

**Figure 12 polymers-12-00967-f012:**
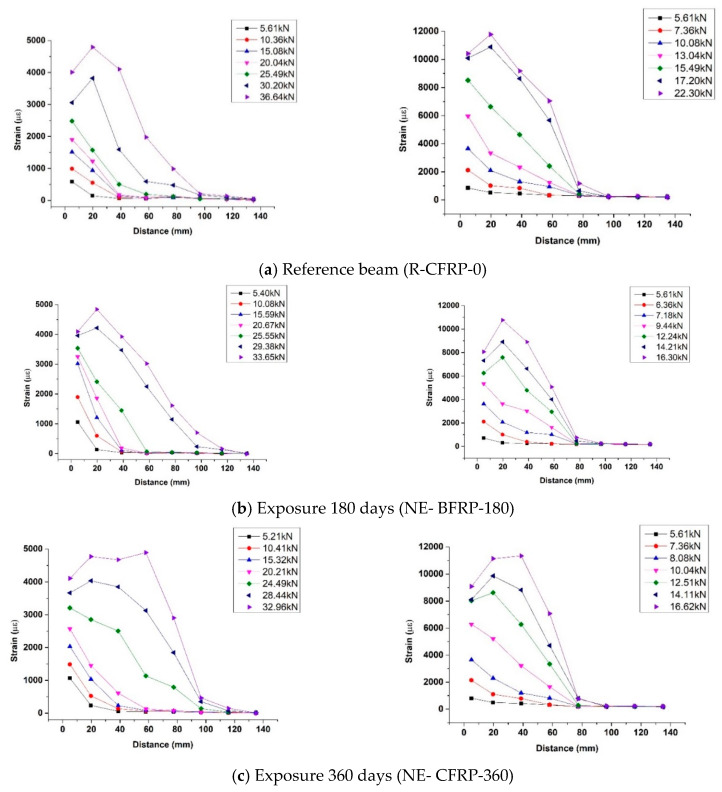
Strains on FRP.

**Figure 13 polymers-12-00967-f013:**
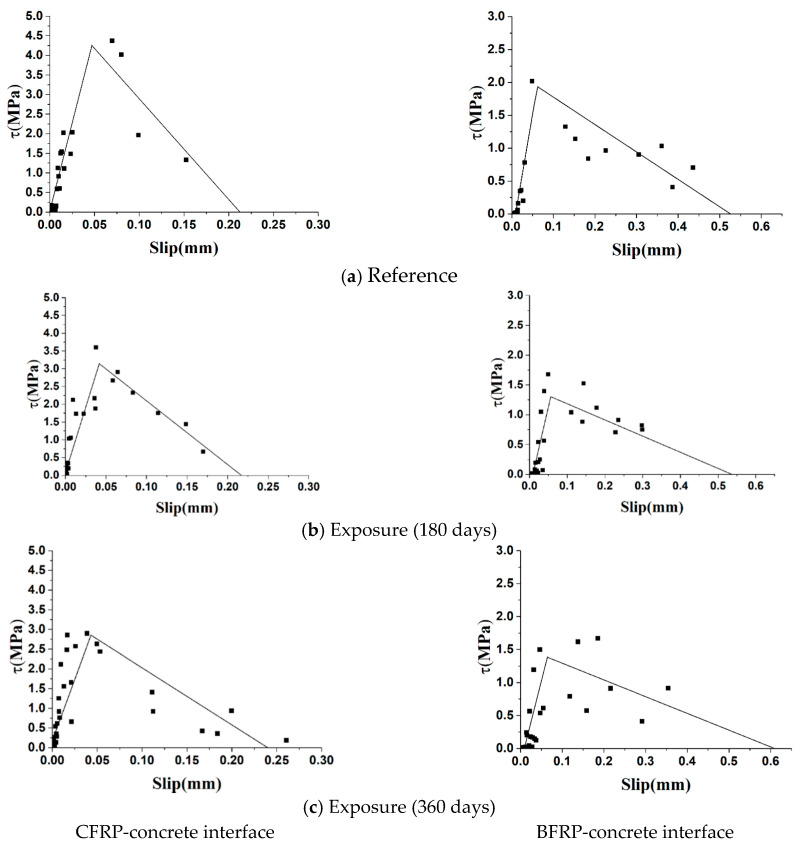
Bond-slip relationships.

**Figure 14 polymers-12-00967-f014:**
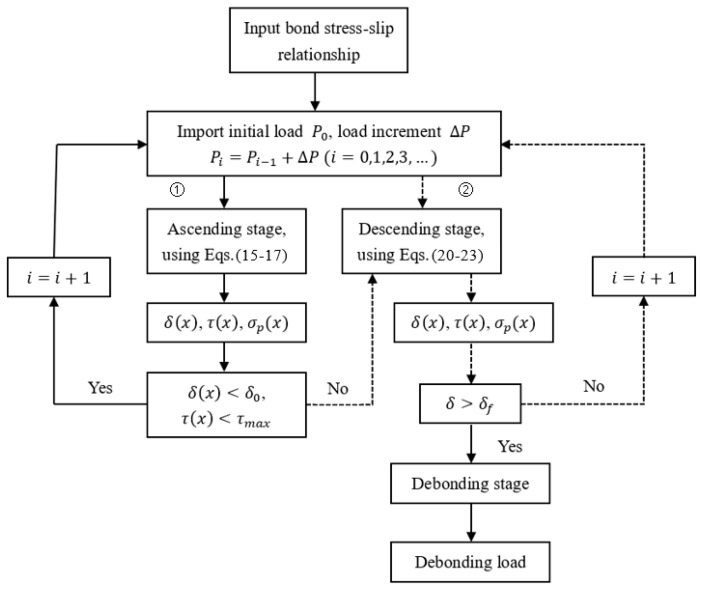
Flow chart for analytical solution.

**Figure 15 polymers-12-00967-f015:**
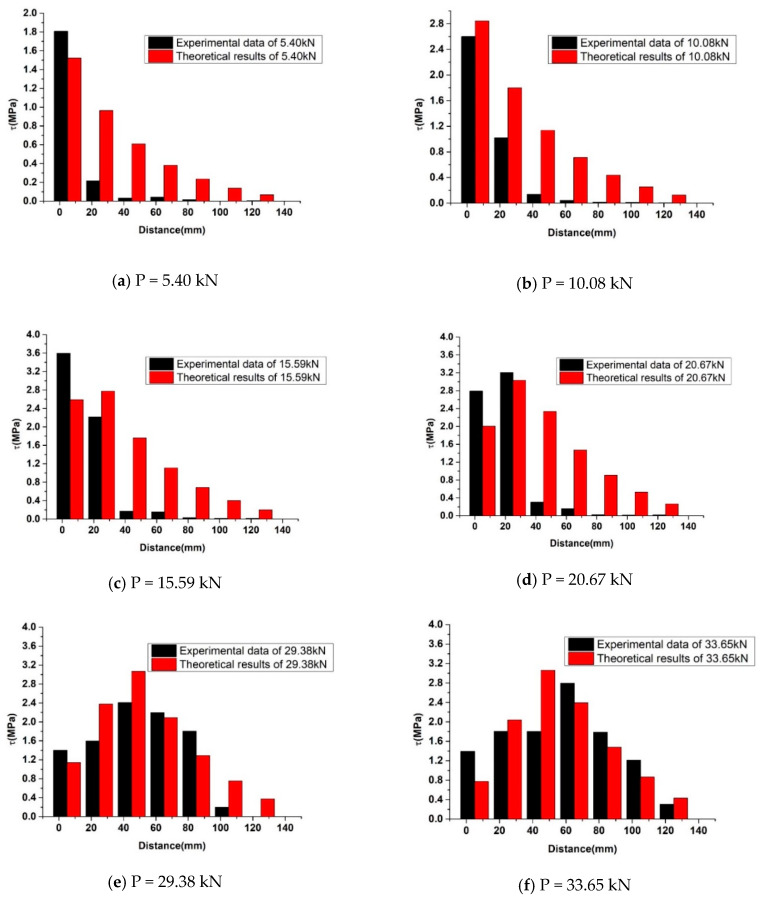
Comparison of analytical and test results of specimen NE-CFRP-180.

**Figure 16 polymers-12-00967-f016:**
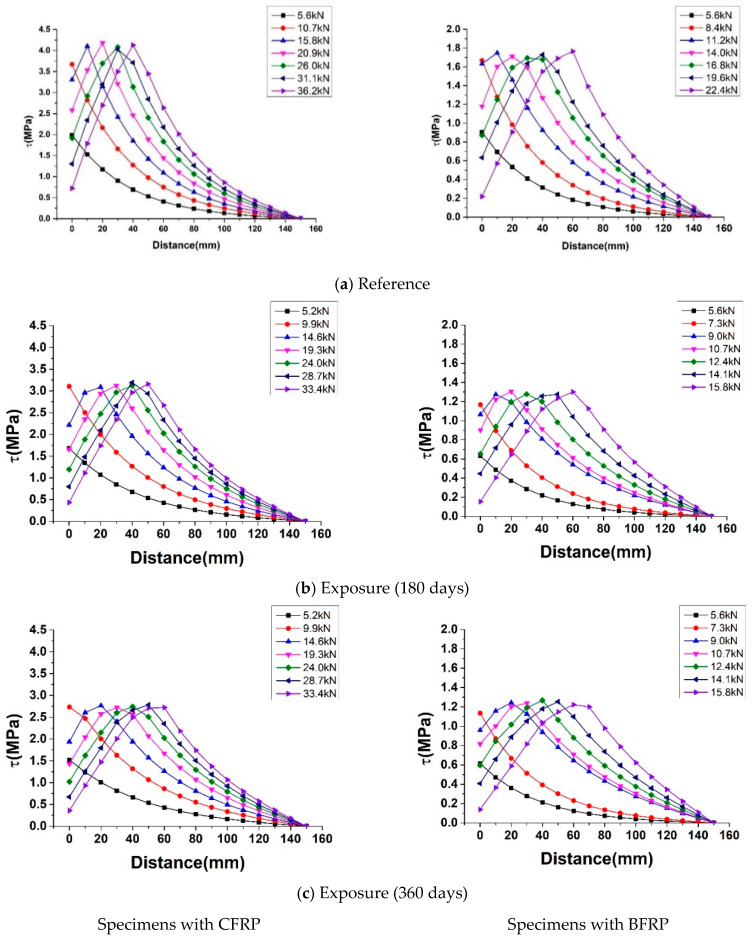
Theoretical results of interfacial shear stress distribution of specimens during the loading.

**Figure 17 polymers-12-00967-f017:**
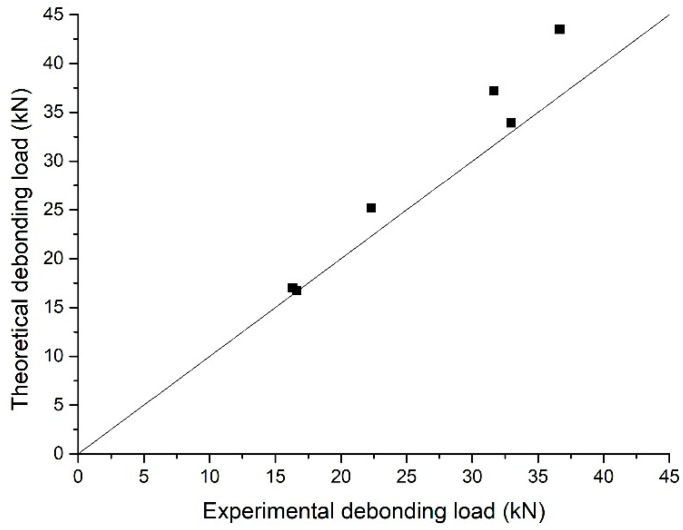
Comparison of theoretical and experimental debonding load.

**Table 1 polymers-12-00967-t001:** Details of material.

Materials	Material Characteristics	Values
Carbon fiber reinforced polymer (CFRP) sheets	Tensile strength (MPa)	4132
	Modulus of elasticity (GPa)	231
	Elongation (%)	1.87
	Thickness (mm)	0.167
Basalt fiber reinforced polymer (BFRP) sheets	Tensile strength (MPa)	2100
	Modulus of elasticity (GPa)	91
	Elongation (%)	2.6
	Thickness (mm)	0.111
Epoxy adhesive	Tensile strength (MPa)	48
	Modulus of elasticity (GPa)	2.378
	Elongation (%)	1.8
Concrete	Compressive strength (MPa)	30.5

**Table 2 polymers-12-00967-t002:** Details of specimens and test results.

Specimens	FRP	Layer	Exposure Period (Days)	Debonding Loads (kN)
R-CFRP-0	CFRP	1	0	36.64
NE- CFRP-180		1	180	31.64
NE- CFRP-360		1	360	32.96
R-BFRP-0	BFRP	1	0	22.30
NE- BFRP-180		1	180	16.31
NE- BFRP-360		1	360	16.62

**Table 3 polymers-12-00967-t003:** Major bond parameters for stress-slip relationships.

Type	Exposure Time (Days)	$\tau_{f}\left(n\right)$ (MPa)	$\delta_{1}\left(n\right)$ (mm)	$\delta_{f}\left(n\right)$ (mm)	$k_{1}\left(n\right)$ (MPa/mm)	$k_{2}\left(n\right)$ (MPa/mm)
CFRP-concrete	0	4.26	0.047	0.212	90.64	25.80
	180	3.11	0.042	0.216	74.05	17.87
	360	2.80	0.045	0.239	62.22	14.43
BFRP-concrete	0	1.94	0.052	0.526	37.31	4.09
	180	1.32	0.056	0.545	23.57	2.70
	360	1.2	0.065	0.610	18.46	2.20

Note: $k_{1}\left(n\right)=\frac{\tau_{f}\left(n\right)}{\delta_{1}\left(n\right)}$, $k_{2}\left(n\right)=\frac{\tau_{f}\left(n\right)}{\delta_{f}\left(n\right)-\delta_{1}\left(n\right)}$.

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
