# Peer review of "Effect of Subtropical Natural Exposure on the Bond Behavior of FRP-Concrete Interface"

_polymers, 2020, doi:10.3390/polym12040967_

Round 1

Reviewer 1 Report

The manuscript is interesting and well organized. Therefore,  I recommend publication of the manuscript after carefully addressing the following comments:

  1. Line 92 – only strains of CFRP were conducted?
  2. Equation 22 is twice.
  3. Line 202 – by Eq. (24) – it is incorrect.
  4. 22 on page 8 should be 23 and last eq. is not seen to the end.
  5. Renumber the equations after number 22.
  6. Line 217 – Fig.6., - delete comma.
  7. Dimensions in Fig.9 do not correspond with dimensions written in the text in page 10.
  8. Line 254 – Twice used “was” in the sentence.
  9. Compressive strength for concrete should be used in Table 1.
  10. Line 303 – Fig.5 - is not right
  11. There is not linear continuity in Figs. 7,8 and 12.
  12. Line 320 – Fig.6 - is not right
  13. Values of delta1 decreased for CFRP in Table 3. You write opposite relation in line 334.
  14. Lines 329, 346, 348, 354 – You write about GFRP material instead BFRP.
  15. Line 354 – mode or model?
  16. You should replace stages 1 and 2 in Fig. 14 and correct the numbers of equations.
  17. The legends in Fig. 15 are not being seen.
  18. 17 is not clear.
  19. Lines 421 and 430 - You write about GFRP material instead BFRP.

Author Response

The authors would like to thank the reviewers and the editors for their constructive comments. All comments have been carefully considered and, wherever appropriate, revisions have been made to the manuscript. Responses to the comments and revisions implemented in the paper are detailed below.

Response to Reviewer #1

Point 1: Line 92 – only strains of CFRP were conducted?

Response 1: In this test, the strains of FRP (CFRP and BFRP) were conducted. “CFRP” in line 92 was replaced by “FRP”.

Point 2: Equation 22 is twice.

Response 2: The formula numbers were all checked and the equations number after number 22 were all revised.

Point 3: Line 202 – by Eq. (24) – it is incorrect.

Response 3: The wrong number“24”in line was revised and replaced by number“22”.

Point 4: 22 on page 8 should be 23 and last eq. is not seen to the end.

Response 4: The formula numbers after number 22 were all checked and revised. The equation 23 was modified.

Point 5: Renumber the equations after number 22.

Response 5: The equations after number 22 were renumbered.

Point 6: Line 217 – Fig.6., - delete comma.

Response 6: The comma in line 217 was deleted.

Point 7: Dimensions in Fig.9 do not correspond with dimensions written in the text in page 10.

Response 7: The wrong dimension “410” in the test was revised as “310”.

Point 8: Line 254 – Twice used “was” in the sentence.

Response 8: Only one “was” used in line 254. The sentence was revised and replaced by “The weight ratio of the main ingredients of concrete was: cement (1.0): water (0.64): sand (2.02): gravel (3.59)”.

Point 9: Compressive strength for concrete should be used in Table 1.

Response 9: “Compressive strength” was used in Table 1.

Point 10: Line 303 – Fig.5 - is not right

Response 10: The wrong number “5” was revised and replaced by “12”.

Point 11: There is not linear continuity in Figs. 7,8 and 12.

Response 11: The dots in Figs. 7, 8 and 12 are experimental value. So there is not linear continuity.

Point 12: Line 320 – Fig.6 - is not right

Response 12: The wrong number “6” was revised and replaced by “13”.

Point 13: Values of delta1 decreased for CFRP in Table 3. You write opposite relation in line 334.

Response 13: The wrong writing “increase” was revised and replaced by “decrease”.

Point 14: Lines 329, 346, 348, 354 – You write about GFRP material instead BFRP.

Response 14: Sorry for my carelessness. I have checked the wrong writing “GFRP” throughout the paper and “GFRP” was replaced by “BFRP”.

Point 15: Line 354 – mode or model?

Response 15: Model

Point 16: You should replace stages 1 and 2 in Fig. 14 and correct the numbers of equations.

Response 16: The wrong writings were revised in Fig.14.

Point 17: The legends in Fig. 15 are not being seen.

Response 17: The Fig.15 was revised and legends can be seen.

Point 18: 17 is not clear.

Response 18: Sorry, I don't understand what "not clear" means.

Point 19: Lines 421 and 430 - You write about GFRP material instead BFRP.

Response 19: Sorry for my carelessness. I have checked the wrong writing “GFRP” throughout the paper and “GFRP” was replaced by “BFRP”.

Reviewer 2 Report

The manuscript is devoted to the bond between CFRP and BFRP/concrete aged interfaces fiber reinforced cementitious matrix (FRCM) and concrete. In the opinion of this reviewer, and despite several drawbacks were found in the text, the work may be worthy of publication in Polymer after major improvements. In order to guide the authors in the improvement of the text, the following comments/questions should be considered:

# Line 58

Although it is impossible to make a complete literature review, the following references should be added due to their relevance to the subject understudy in the manuscript:

https://doi.org/10.1061/(ASCE)MT.1943-5533.0001008

https://www.concrete.org/publications/internationalconcreteabstractsportal.aspx?m=details&id=51686153

https://doi.org/10.1016/j.compstruct.2010.05.018

https://doi.org/10.1061/(ASCE)CC.1943-5614.0000600

# Line 82

The advantages and disadvantages should be clear for the readers and therefore, must be enumerated.

# Line 89

The model shown by the authors does not consider the bending moment. Please clarify why not?

# Lines 101-104

The same comment made before can be applied here. The differential equation to be derived does not consider the shear nor the bending effects on the bond between the FRP and the concrete. Further explanations should be added in order to not miss lead the readers.

# Line 107

This reviewer does not agree with this sentence as explained before.

# Line 135

Instead of "four types of available bond-slip models of FRP-concrete interface", and to the best knowledge of this reviewer, a fifth model can be found in:

https://doi.org/10.1061/(ASCE)CC.1943-5614.0000956

https://doi.org/10.1016/j.compstruct.2018.12.053

a six one in:

https://doi.org/10.1061/(ASCE)1090-0268(2005)9:1(52)

and a seventh in:

https://doi.org/10.1016/j.engstruct.2011.02.036

# Lines 155 and 158

Reference already mentioned before (https://doi.org/10.1061/(ASCE)CC.1943-5614.0000956 and https://doi.org/10.1016/j.compstruct.2018.12.053), should be added here.

# Line 203, Eq. 22

This equation is not complete in the manuscript. Please check it.

# Line 209, Figure 5

The transition between the elastic and the softening stage is not well represented. As it is in this figure, it can be seen that at maximum bond stress, tauf, the function is continuous and with the bi-linear bond-slip model a discontinuity should be obtained at maximum bond stress. See for instance references https://doi.org/10.1016/j.engstruct.2016.07.030 or https://doi.org/10.1016/j.engstruct.2003.11.006. So, this figure is just a scheme badly made or are the results already obtained from the analytical model? Please clarify.

# Line 207

The expression for Pu is missing.

# Line 225, Figure 6

This figure is conceptually wrong. First, the bond stress at the FRP free end is not always zero. Second, the softening stage is not represented. And third, there is no reason to change (?) the y-axis at every load increase. Therefore, the analytical model derived in this section should be revised. Please correct and comment.

# Line 236, Figs. 7, 8

The legends are cut in both figures (check all figures in the text also). This reviewer did not understand what the authors meant with "bridge". Please clarify.

# Line 249, Fig. 9

The shear span is quite meaningful and the external loads will directly go to both supports and compromising, therefore, the final results. Please comment.

# Line 256

How many tests were made to characterize the concrete? What is the corresponding coefficient of variation (CoV)? Please clarify.

# Line 258, Table 1

  1. a) The mechanical properties of the CFRP and BFRP were experimentally obtained or the values given by the manufacturer were assumed? If they were experimentally obtained, refer to those as mean values. The CoV is also an important value that should be reported as well. Please clarify.
  2. b) Why the authors did not consider the same aging periods for concrete and CFRP/BFRP? Please clarify.

# Line 286

How many tests were carried out for each condition? Can't be only one in a total of 6 specimens because for a complicated subject the probability to get scattered results is quite high. Therefore, the debonding loads represent an average value, right? Please clarify.

# Lines 316-317

What is the basis of this sentence? How we can confirm this by observing Fig. 12? Please clarify.

# Line 318

How did the author calculate the bond stresses and the slips in this subsection 4.4? Please clarify.

# Line 325, Table 3

  1. a) As already mentioned before, the results can be obtained from one single specimen for each condition and therefore, the results shown in this table should be average ones. Please clarify.
  2. b) How the authors obtained these values? Any minimization process was carried out? Any other strategy? Please clarify.

# Line 326

Instead of "Not" should be "Note".

# Line 392, Fig. 16

  1. a) The same x-axis and y-axis should be used in the graphs pf this figure. Please adopt this in all figures.
  2. b) The bond stress distributions shown in this figure are not correct because at the free end it is imposed that the bond stress is always 0 and, consequently, the slip is 0 also. This is the case when we have an anchorage that prevents the slip which is not the case under study. Please revised this.

# Line 404

The load-slip curve should be reported and compared with the experimental results.

# Line 413, Fig. 17

Only one data from each case experimentally obtained is shown. All the experimental data should be reported. Please keep in mind that only one test is hard to accept due to its very low statistical meaning.

# Line 414

Conclusions should be improved in accordance with the comments made and corrections introduced into the text.

Author Response

The authors would like to thank the reviewers and the editors for their constructive comments. All comments have been carefully considered and, wherever appropriate, revisions have been made to the manuscript. Responses to the comments and revisions implemented in the paper are detailed below.

Response to Reviewer #2

Point 1: # Line 58

Although it is impossible to make a complete literature review, the following references should be added due to their relevance to the subject understudy in the manuscript:

https://doi.org/10.1061/(ASCE)MT.1943-5533.0001008

https://www.concrete.org/publications/internationalconcreteabstractsportal.aspx?m=details&id=51686153

https://doi.org/10.1016/j.compstruct.2010.05.018

https://doi.org/10.1061/(ASCE)CC.1943-5614.0000600

Response 1: These studies were added in references [10-12].

Point 2: # Line 82

The advantages and disadvantages should be clear for the readers and therefore, must be enumerated.

Response 2: The expression of advantages and disadvantages “The advantage of single-shear test is easy operation. However, it is rather difficult to keep loading end horizontal, which may introduce to eccentric loading. Although the double-shear test can overcome the shortcoming of eccentric tension, it cannot simulate the actual situation of concrete structure strengthened with FRP because most concrete structures were in bending” were added in line 83-86.

Point 3: # Line 89

The model shown by the authors does not consider the bending moment. Please clarify why not?

Response 3: This study mainly focused on the bond slip of FRP-concrete interface. Based on the force analysis of the specimens designed in this paper, between the two load points, the shear force on the concrete is 0, the bending moment of concrete is very small, and shear stress of FRP-concrete interface is maximum. The tensile stress of FRP was caused by the bending moment applied on the specimens. The behavior of FRP-concrete interface was analyzed under the tensile stress of FRP.

Meanwhile, in our previous studies on the FRP-concrete interface of whole RC beams strengthened with FRP, a model was established (as shown in the figure below) to find the maximum shear stress and normal stress, in which the influences of shear force and bending moment were considered. The results showed that the normal stress is much smaller than the shear stress. Hence, to reduce the complexity of the theoretical model, the effect of normal stress on the mechanical properties of FRP-concrete interface is not considered.

In view of the above, the shear force and bending moment are not considered in this model.

Point 4: # Lines 101-104

The same comment made before can be applied here. The differential equation to be derived does not consider the shear nor the bending effects on the bond between the FRP and the concrete. Further explanations should be added in order to not miss lead the readers.

Response 4: The explaination was added at the beginning of section 2.1.

Point 5: # Line 107

This reviewer does not agree with this sentence as explained before.

Response 5: Because the shear force and bending moment were not considered in this study, the theoretical analyses (the bond-slips of FRP-concrete interface under different subtropical natural exposure were substituted into structure to find the ultimate load) can be used in single and double specimens. To overcome the misunderstanding, the line 107 was deleted.

Point 6: # Line 135

Instead of "four types of available bond-slip models of FRP-concrete interface", and to the best knowledge of this reviewer, a fifth model can be found in:

https://doi.org/10.1061/(ASCE)CC.1943-5614.0000956

https://doi.org/10.1016/j.compstruct.2018.12.053

a six one in:

https://doi.org/10.1061/(ASCE)1090-0268(2005)9:1(52)

and a seventh in:

https://doi.org/10.1016/j.engstruct.2011.02.036

Response 6: Three other models proposed by the reviewers and corresponding references were added.

Point 7: # Lines 155 and 158

Reference already mentioned before (https://doi.org/10.1061/(ASCE)CC.1943-5614.0000956 and https://doi.org/10.1016/j.compstruct.2018.12.053), should be added here.

Response 7: Two references mentioned by reviewer were added.

Point 8: # Line 203, Eq. 22

This equation is not complete in the manuscript. Please check it.

Response 8: The equation 23 was modified.

Point 9: # Line 209, Figure 5

The transition between the elastic and the softening stage is not well represented. As it is in this figure, it can be seen that at maximum bond stress, tauf, the function is continuous and with the bi-linear bond-slip model a discontinuity should be obtained at maximum bond stress. See for instance references https://doi.org/10.1016/j.engstruct.2016.07.030 or https://doi.org/10.1016/j.engstruct.2003.11.006. So, this figure is just a scheme badly made or are the results already obtained from the analytical model? Please clarify.

Response 9: Fig.4- Fig.6 are all schematic diagram made to show the regular of shear stress distribution. The explanations were added in line 200,230,269.

Point 10: # Line 207

The expression for Pu is missing.

Response 10: The expression for is debonding load and the criterion of debonding failure was that the slip reach the maximum slip was added.

Point 11: # Line 225, Figure 6

This figure is conceptually wrong. First, the bond stress at the FRP free end is not always zero. Second, the softening stage is not represented. And third, there is no reason to change (?) the y-axis at every load increase. Therefore, the analytical model derived in this section should be revised. Please correct and comment.

Response 11: Thanks to the reviewer for pointing out the mistake. The Fig.6 was revised.

Point 12: # Line 236, Figs. 7, 8

The legends are cut in both figures (check all figures in the text also). This reviewer did not understand what the authors meant with "bridge". Please clarify.

Response 12: All figures were checked and the same mistake in Fig7, 8,12,15,16 were revised. Our group measured the temperature and humidity on real bridge. The "bridge" in Fig.7 and Fig.8 means the concrete structures. I explained it in line 304.

Point 13: # Line 249, Fig. 9

The shear span is quite meaningful and the external loads will directly go to both supports and compromising, therefore, the final results. Please comment.

Response 13: The shear span is quite meaningful for RC beam strengthened with FRP. But in this paper, the shear span only influence the tensile force applied on FRP.

Point 14: # Line 256

How many tests were made to characterize the concrete? What is the corresponding coefficient of variation (CoV)? Please clarify.

Response 14: Three tests were made to characterize the concrete. Explanation “Three cylindrical test blocks were tested” was added in line 357.The corresponding coefficient of variation was not measured in this test.

Point 15: # Line 258, Table 1

  1. a) The mechanical properties of the CFRP and BFRP were experimentally obtained or the values given by the manufacturer were assumed? If they were experimentally obtained, refer to those as mean values. The CoV is also an important value that should be reported as well. Please clarify.
  2. b) Why the authors did not consider the same aging periods for concrete and CFRP/BFRP? Please clarify.

Response 15: a) The mechanical properties of the CFRP and BFRP were given by the manufacturer and the explaination was added in line 354. b) During the test, the same aging periods for concrete, resin epoxy, and CFRP/BFRP were conducted, but the results were not written in this paper. In this paper, the main purpose was focused on bond slip of FRP-concrete interface under subtropical natural exposure. So the test results corresponding to single material were not involved.

Point 16: # Line 286

How many tests were carried out for each condition? Can't be only one in a total of 6 specimens because for a complicated subject the probability to get scattered results is quite high. Therefore, the debonding loads represent an average value, right? Please clarify.

Response 16: In previous experiments, many specimens were tested with many variables, such as types of concrete, types of FRP, types of aging environments and so on. Only 6 specimens were successfully tested under subtropical natural exposure. From the test result, the degradation of FRP-concrete was obvious. Every specimen has one debonding load.

Point 17: # Lines 316-317

What is the basis of this sentence? How we can confirm this by observing Fig. 12? Please clarify.

Response 17: From Fig.12, we can see that the tendency of peak strain moving backwards with the exposure period increases. The phenomenon that the strain of FRP began to increase under the same load and the peak strain moving backwards at peak load showed the degradation of bond behavior.

Point 18: # Line 318

How did the author calculate the bond stresses and the slips in this subsection 4.4? Please clarify.

Response 18: The average bond stress between two strain gauges can be obtained from the following equation:

where is the elastic modulus of FRP; is the height of FRP; and are strain values of two adjacent strain gauges; is the distance of two adjacent strain gauges.

To calculate the slip , we assume:

  • The strain is linear distribution between two adjacent strain gauges.
  • The slip at free end of FRP is 0.

Thus, the slip at point can be obtained from the following equation:

Some details were added in line 435-439.

Point 19: # Line 325, Table 3

  1. a) As already mentioned before, the results can be obtained from one single specimen for each condition and therefore, the results shown in this table should be average ones. Please clarify.
  2. b) How the authors obtained these values? Any minimization process was carried out? Any other strategy? Please clarify.

Response 19: a) 6 specimens were successfully tested. The results shown in Table 3 are values from the double lines in Fig.13. The double line of each specimen was fitted by shear stress and slip of each specimen. The explaination was shown in line 440. b) These values in Table 3 were obtained from the bond-slip relationships in Fig.13.

Point 20: # Line 326

Instead of "Not" should be "Note".

Response 20: The mistake was revised.

Point 21: # Line 392, Fig. 16

  1. a) The same x-axis and y-axis should be used in the graphs pf this figure. Please adopt this in all figures.
  2. b) The bond stress distributions shown in this figure are not correct because at the free end it is imposed that the bond stress is always 0 and, consequently, the slip is 0 also. This is the case when we have an anchorage that prevents the slip which is not the case under study. Please revised this.

Response 21: a) The y-axis in Fig.16 were modified. b) Because the bonding length in this study was larger than the effective bonding length, the bond stress at free end is zero. The test result showed the strain on FRP and the shear stress were almost zero.

Point 22: # Line 404

The load-slip curve should be reported and compared with the experimental results.

Response 22: The bilinear bond stress-slip relationships for different exposure periods were obtained from the experimental results. Based on these bond-slip relationships, the full-range behavior of shear stress along the bond length and debonding load can be obtained through the analytical solution.

Point 23: # Line 413, Fig. 17

Only one data from each case experimentally obtained is shown. All the experimental data should be reported. Please keep in mind that only one test is hard to accept due to its very low statistical meaning.

Response 23: The results from the tests include ultimate loads, failure modes, and strains of FRP under different loads. All the experimental data are reported in section 4. I thought there was no experimental data missing.

Round 2

Reviewer 2 Report

In the opinion of this reviewer, the authors have changed the manuscript according to the comments and/or suggestions raised and the drawbacks pointed out to the m/s before were all corrected. Therefore, the m/s is now worthy of publication in Polymers.